# Processing and Characterisation of Alumina/Eucryptite Nanostructured Composites

**DOI:** 10.3390/ma18030671

**Published:** 2025-02-03

**Authors:** Jordana Mariot Inocente, Renata Bochanoski da Costa, Ana Sônia Mattos, Carmen Alcázar, Amparo Borrell, Rodrigo Moreno, Sabrina Arcaro, Oscar Rubem Klegues Montedo

**Affiliations:** 1Laboratório de Cerâmica Técnica (CerTec), Programa de Pós-Graduação em Ciência e Engenharia de Materiais (PPGGEM), Universidade do Extremo Sul Catarinense (UNESC), Criciuma 88806–000, Brazil; jordanainocente@gmail.com (J.M.I.); renatacosta@unesc.net (R.B.d.C.);; 2Institute of Ceramics and Glass, CSIC, 28049 Madrid, Spain; 3Escuela Técnica Superior de Ingeniería Industrial, Universitat Politècnica de València (UPV), 46022 Valencia, Spain; aborrell@upv.es

**Keywords:** nanostructured composite, alumina, eucryptite, fracture toughness

## Abstract

Alumina is one of the most studied and used ceramic materials, but increasing its fracture toughness is still a challenge for many specific impact applications. Adding a second phase with a low coefficient of thermal expansion (CTE) to an alumina matrix can enhance the matrix’s mechanical properties, reduce its sintering temperature, and increase its toughness by generating compressive stresses on the alumina particle surface. In this study, nanostructured alumina/eucryptite composites were prepared to achieve enhanced toughness. First, eucryptite (Li_2_O·Al_2_O_3_·2SiO_2_) nanoparticles were successfully synthesised via colloidal heterocoagulation. These nanoparticles were then used to reinforce alumina matrices through slip casting followed by conventional sintering. Complete crystallisation of eucryptite was achieved at 850 °C with a CTE of 0.46 × 10 ^−6^ °C ^−^¹. Transmission electron microscopy analysis revealed that the average particle size was 28.5 ± 14.5 nm. To achieve a relative density of 95.3%, the composite containing 5 vol.% eucryptite required sintering for 1 h at 1400 °C whereas pure alumina required 2 h at 1600 °C. This reduction in sintering temperature (by up to 200 °C) helped to improve the fracture toughness, with the alumina grain size decreasing from 2.3 to 0.9 µm. The advantages of the new composite are the more economically viable and environmentally friendly way of producing the lithium aluminosilicate nanoparticles, compared to the production of ceramic frits at high temperatures (~1500 °C).

## 1. Introduction

Over the last 20 years, complex structures combining different classes of materials have been designed to improve the mechanical, thermal, and chemical properties of ceramics [1]. Adding a second phase to the alumina matrix at concentrations below 10 vol.% may enhance the mechanical properties of the material. In addition to lowering the sintering temperature, particles of the second phase often contribute to material toughening by increasing the energy released during crack propagation [2]. Other explored toughening mechanisms include modification of the fracture mode due to crack deflection between phases [3,4] and the generation of compressive stresses on the alumina particle surface [5]. Karandikar et al. [3] have presented an extensive review about ceramics for armour applications, while Montedo et al. [4] have described the sintering behaviour of alumina containing up to 21 vol.% of the 11.6Li_2_O·16.8ZrO_2_·68.2SiO_2_·3.4Al_2_O_3_ (LZSA) glass–ceramic for high wear resistance applications.

A previous study on SiC-reinforced Al_2_O_3_ composites fabricated by hot pressing concluded that the SiC particles can control grain growth, presumably by inhibiting diffusion at the grain boundaries [6]. Improvements in mechanical properties such as fracture toughness (K_IC_) have been attributed to a crack deflection mechanism resulting from a fine and uniform microstructure [6]. Montedo et al. [5,7] prepared glass–ceramic materials by adding 7, 15, and 21 vol.% of 11.6Li_2_O-16.8ZrO_2_-68.2SiO_2_-3.4Al_2_O_3_ to submicrometric alumina. The addition of the glass–ceramic has suppressed the grain growth of alumina contributing to an increased fracture toughness [5], while X-ray diffraction (XRD) analysis revealed that compressive residual stresses were developed in this composite, increasing the fracture toughness from 2.2 MPa·m^0.5^ for pure alumina to 4.9 MPa·m^0.5^ for the composite with 21 vol.% LZSA and sintered at 1470 °C for 40 min, due to a dominant β-spodumene crystalline phase with a low coefficient of thermal expansion (CTE) [7]. However, the presence of a residual glassy phase and other phases with higher CTEs may have placed an upper limit on the residual compressive stress. Although the addition of a LZSA glass–ceramic has improved the mechanical properties of alumina, the production of ceramic frits requires high temperatures (~1500 °C), representing a disadvantage from an economic and environmental point of view.

Alternatively, a low-CTE crystalline phase such as β-spodumene (Li_2_O·Al_2_O_3_·4SiO_2_, 0.2 × 10^−6^ °C^−1^) could be added in place of the β-spodumene-based glass–ceramic. Adding a crystalline phase with an even lower CTE value, such as eucryptite (Li_2_O·Al_2_O_3_·2SiO_2_, –0.9 × 10^−6^ °C^−1^), may also create compressive residual stresses on alumina particles and further improve the fracture toughness.

Researchers have explored various kinetic and thermodynamic parameters, including the nucleation and crystallisation temperatures, to obtain crystalline phases of the lithium aluminosilicate family through the fusion of compositions, producing frits and subsequently low-CTE glass–ceramics [8]. Starting from micrometre-sized precursors, Wang et al. [9] obtained β-spodumene and eucryptite (CTE = −0.9 × 10^−6^ °C^−1^ [10]) via solid-state reactions at 1200 °C. However, there are no reports on producing eucryptite nanoparticles through colloidal processing. There is also a lack of knowledge on using eucryptite nanoparticles to increase the fracture toughness of alumina by generating a compressive stress at the alumina/eucryptite interface due to a difference in the CTE between these crystalline phases.

Based on the above background, this study aims to prepare eucryptite nanoparticles by colloidal synthesis via heterocoagulation, further preparation of nanostructured alumina/eucryptite composites, and final evaluation of the effect of eucryptite particles on the toughening of alumina.

## 2. Materials and Methods

Eucryptite nanoparticles were prepared using the following raw materials: commercial alumina in aqueous suspension (Aerodisp W630-Evonik, Essen, Germany, with solid content of 32 wt.%, average particle size of 25 nm, pH 4), colloidal silica (Ludox TM 40−Aldrich, Darmstadt, Germany, with solid content of 40 wt.%, average particle size of 24 nm, pH 9), Li_2_CO_3_ (Synth, 99% purity, Brazil), and glacial acetic acid (Applichem-Panreac, Barcelona, Spain, with 99.5% purity).

Alumina/eucryptite composites were prepared using the synthesised eucryptite nanoparticles and the following raw materials: commercial alumina (CT3000SG, Almatis, Frankfurt am Main, Germany, 99.8% purity, average particle size of 0.5 μm), a poly(acrylic) acid-based polyelectrolyte dispersant (Duramax™ D-3005, 35%, Dow Chemicals, Russellville, AR, USA purity), and deionised water.

### 2.1. Preparation and Characterisation of Eucryptite Nanoparticles

Based on previous studies [11,12,13], we used the heterocoagulation route to prepare nanometre-sized eucryptite particles. When particles with a high surface area are very close, due to the forces of attraction, an external force, such as heat treatment, promotes the formation of new crystalline phases [14]. Figure 1 shows the flowchart of the experimental steps of the work, while the synthesis stages are shown in Figure 2.

First, to obtain 5 g of the final eucryptite phase, a solution containing deionised water (50 mL), lithium carbonate (0.70 cm³), and glacial acetic acid (2.3 mL) was prepared. A glacial acetic acid solution (5 mol/L) was used as the solvent to ensure a sufficient lithium ion concentration and decompose Li_2_CO_3_ into Li_2_O and CO_2_ (for the reaction see Ref. [13]). The solution was stirred for 30 min at room temperature (25 °C) using a magnetic stirrer (Fisotom 753, Brazil), and then the pH was adjusted to 11. The aqueous suspension of Al_2_O_3_ nanoparticles (5.14 mL) were slowly added, and the resulting suspension was stirred constantly for another 30 min. Next, SiO_2_ nanoparticles were slowly added (4.85 mL). The mixed suspension was magnetically stirred for 1 h and further subjected to ultrasonic dispersion (Hielscher Ultrasonics UP400S, Teltow, Germany) for 1 min to evenly disperse the nanoparticles.

After adjusting the pH to 6.6, the suspension was placed in a volumetric flask connected to a rotary evaporator (RV 10 Basic, IKA, Staufen, Germany; 100 rpm), frozen in a liquid nitrogen bath, and freeze-dried at 5 Pa and –50 °C for 24 h (Cryodos 50, Telstar, Barcelona, Spain).

A freeze-dried sample (20 mg) was placed in a platinum crucible for thermogravimetric analysis and differential scanning calorimetry (DSC-TG), using a Setaram Setsys Evolution system (Caluire, France) in an oxidising atmosphere (air). The heating rate was 10 °C/min and the temperature range was 25 to 1300 °C. After comparing the DSC-TG data to the literature, the obtained powders were heated at 700, 750, 800, 850, and 900 °C in order to identify the temperature that maximises the formation of eucryptite. Specifically, heating was carried out in a Nabertherm L5/11/P330 (Bremen, Germany) furnace in air atmosphere at a rate of 10 °C/min. After reaching 500 °C, the sample was held for 1 h to eliminate the organic additives. Upon reaching the target temperature, the sample was held for another 1 h.

Crystalline phases in the sample after thermal treatment were determined by XRD using a Bruker D8 Advance diffractometer (Karlsruhe, Germany) with a Lynxeye detector and a germanium monochromator. The X-ray came from a monochromatic CuKα1 source (40 kV, 30 mA, λ = 1.5406 Å), and data were collected in the 2θ range of 5–70° with a step size of 0.02° and a counting time of 1 s/step. Rietveld refinement was performed using the X’Pert HighScore Plus^®^ software (version 3.0), utilising the Inorganic Crystal Structure Database (ICSD) provided by the Crystalline Structures Database (BDEC). From the obtained diffractograms, the crystallite size was determined using the Debye–Scherrer equation [15]:D = K·λ / (β·cos θ)(1)
where D is the apparent particle size in nm, K is the Scherrer constant and typically assumed to be 0.9, λ = 1.5406 Å is the X-ray wavelength, β is the full width at half maximum (FWHM) after subtracting instrumental broadening, and θ is the Bragg angle. Equation (1) is limited to crystallite sizes smaller than 100 nm, and its optimal application range is 2θ = 30°–50° [15].

CTE values in the temperature range of 125–450 °C were determined using a Netzsch 402 PC1 (Selb, Germnay) dilatometer equipped with a silica sample holder and push rod at a heating rate of 5 °C/min up to 500 °C. The true density of the powder was measured using a helium gas pycnometer (UltraPyc 1200e, Anton Paar, Graz, Austria) operated with an integrated system for equilibrium and volume displacement measurements. Microstructural analysis of the nanoparticles was carried out using a transmission electron microscope (TEM; Jeol JEM-1400, 120 keV, Akishima, Japan) coupled with an energy-dispersive X-ray spectroscopy (EDS) microprobe.

The specific surface area was determined based on the Brunauer–Emmett–Teller (BET) equation using nitrogen adsorption (Quantachrome Monosorb, Boynton Beach, FL, USA). First, the sample was degassed at 80 °C for 6 h, and then nitrogen adsorption–desorption was carried out to collect the isotherm (10 data points for adsorption and 10 data points for desorption).

To validate the heterocoagulation process, zeta potentials were measured at different pH for the precursor materials (alumina and colloidal silica) using laser Doppler velocimetry (Zetasizer NanoZS, Malvern, UK). The pH was controlled using 10^−1^ M solutions of HCl and KOH, and 10^−1^ M solution of KCl was used as the inert electrolyte.

### 2.2. Preparation and Characterisation of Alumina/Eucryptite Composites

Previous studies [4,7] showed that the content of β-spodumene in glass–ceramics should be kept at no more than 5 vol.% to avoid degrading the mechanical properties. Therefore, alumina composites containing 5 vol.% eucryptite nanoparticles were prepared and compared to pure alumina.

The first step in producing ceramics using the slip casting process is to prepare a stable, well-dispersed suspension of raw materials with suitable rheological properties. Because nanoparticles are thermodynamically unstable and tend to agglomerate [16], dispersants are often added to improve the homogeneity of ceramic powders in liquid media [17]. Moreover, probe sonication can quickly and effectively deagglomerate and disperse nanoparticles in liquids [18]. In this study, we used a polymeric dispersant (Duramax™ D-3005, Dow Chemicals, MI, USA; 0.5 wt.% on a dry basis) and also sonicated the dispersion for different times (0, 1, 2, and 3 min).

Rheological analysis was carried out at 25 °C using an RS50 rheometer (Thermo Haake, Karlsruhe, Germany) with a double-cone/plate sensor configuration (DC60/2 Ti, Thermo Haake, Karlsruhe, Germany) and a sample volume of ~5 mL. The viscous flow behaviour was evaluated using controlled rate (CR) tests employing a three-stage measurement program. First, the shear rate was increased linearly from 0 to 1000 s^−1^ in 300 s, the maximum shear rate was maintained for 60 s, and then the shear rate was reduced linearly from 1000 to 0 s^−1^ in 300 s. The rheological properties of suspensions depend strongly on the solid content: a higher solid content means a smaller particle–particle distance, a stronger interparticle force, and a higher viscosity [19]. Based on the obtained rheological data and preliminary test results, we decided to use suspensions with a total solid content of 40 vol.% (with eucryptite and alumina in 5:95 *v*/*v*) and a total volume of 100 cm³ to prepare the composites.

Specifically, 60 mL of deionised water was placed in an Erlenmeyer flask. Under continuous mechanical agitation, 0.5 wt.% of the polymer dispersant was added, followed by the eucryptite nanoparticles, and then alumina powder. Finally, ultrasound was applied for 0, 1, 2, and 3 min before the rheological study.

For slip casting, the suspensions were poured into cylindrical plastic rings (diameter: 25 mm, height: 20 mm) placed over a permeable base made of plaster of Paris. The specimens were left to dry at room temperature (25 °C) for 48 h, removed from the mould, and dried in a laboratory oven at 100 °C for 24 h. The dried samples were then sintered.

A 5 mm × 5 mm × 5 mm sample was cut and tested using a Setsys vertical dilatometer (Setaram, Caliure, France) to obtain linear shrinkage curves as a function of temperature and sintering time. The test consisted of two parts. First, the sample was heated at 10 °C/min up to 1600 °C to identify the temperature with the highest shrinkage rate. Second, the sample was heated at 10 °C/min to the identified temperature with the highest shrinkage rate, and then held for 1 h to determine the time required to achieve maximum densification. This way, the optimal heat-treatment temperature and duration were determined for each composition (Table 1).

Using the conditions in Table 1, cylindrical (discs) green bodies were sintered in a furnace (L5/11/P330, Nabertherm, Bremen, Germany), and then their mechanical and microstructural characteristics were analysed. To eliminate organic components, all samples were preheated slowly (2 °C/min) to 500 °C and held for 2 h, before ramping at a rate of 10 °C/min to the temperatures shown in Table 1.

The true density (ρ_true_) of powder samples was determined using a helium pycnometer (Ultrapyc 5000, Anton Paar, Graz, Austria). The apparent density (ρ_ap_) of sintered samples was determined using the Archimedes principle by immersion in water at 25 °C, employing a density measuring device equipped with a balance (Shimadzu AX200, Kyoto, Japan, ± 0.0001 g). Porosity (P) was calculated from ρ_ap_ and ρ_true_ using Equation (2) [7]:(2)$$P=\left(1-\frac{\rho_{ap}}{\rho_{true}}\right)\times 100\%$$

The hardness and modulus of elasticity of specimens were assessed using a microindentation technique. A small part of the sample was embedded in epoxy resin and polished. The mechanical properties were calculated from the indentation marks, and the stresses produced in these marks were observed using confocal Raman microscopy both on the surface and at a depth. Microindentation was carried out using a CETR/Bruker Apex microtribometer (Karlsruhe, Germany) with a Vickers indenter and calibrated with a fused silica standard. The maximum applied load was 15 N, the loading rate was 1.5 N/s, and the maximum load-holding time was 10 s. The measurements were performed in accordance with the ISO 14577-1 standard. The surfaces were further observed using field-emission scanning electron microscopy (SEM; Hitachi FEG-SEM S-4800, Marunouchi, Japan) [20].

The modulus of elasticity was calculated using the Oliver–Pharr method based on the contact area of the indenter (A_c_), which depends on the depth of contact at full load (h_c_) as A_c_ = 24.56 h_c_^2^ [21]. The fracture toughness (K_IC_) was calculated using the experimental equation proposed in [22], by relating the lengths of the cracks growing at the corners of the Vickers indentation under an applied load to the fracture toughness of the material.

Flexural strength was assessed using the “ball on three balls” biaxial test, which is less sensitive to variations in the flatness of the disk, because the friction between the sample and the load ball is minimal [23,24]. This test requires precise knowledge of the relationship between the applied load and maximum tensile stress. The mathematical expression to calculate flexural strength was proposed by Kirstein and Woolley [25] and Vitman and Pukh [26], and it is independent of the number of support balls. From the obtained results, the maximum stress in the disks had an error less than 2%, although this error can be significantly higher for thinner disks [27]. The bending tests were conducted at room temperature on parallel planar surfaces using five specimens for each measurement, according to the standardised ASTM F394-78 method [28]. A Shimadzu (Kyoto, Japan) AG-X PLUS 100KN machine was used with a load displacement speed of 0.012 mm/s.

Based on the measured fracture toughness and mechanical resistance to biaxial bending, it was possible to calculate the size of the natural Griffith defect. Griffith defects refer to small cracks or defects in ceramic materials, and they are important in fracture mechanics [29]. According to fracture mechanics, the size of a natural defect can be calculated using Equation (3):(3)$$\sigma =\frac{K_{IC}}{Y\cdot {ɑ}^{1/2}}$$
where σ is the flexural strength, K_IC_ is the fracture toughness, ɑ is the size of the natural Griffith defect, and Y is a calibration factor.

The microstructure of the materials, their typical defects, grain size, and fracture mode were analysed using SEM (Zeiss EVO MA10, Oberkochen, Germany) with an attached EDS microprobe. This analysis was performed on the parts used in the indentation hardness test. Specifically, samples with an already polished surface were demoulded from epoxy resin and subjected to a thermal attack to reveal the grain contours of alumina. The thermal attack was carried out for 15 min at 100 °C below the sample sintering temperature, and then the furnace was opened to allow rapid cooling. Finally, the samples were sputtered in vacuum with a thin layer of gold in a Quorum QR 150ES metalliser (Laughton, UK) before the SEM-EDS observations. The average grain size was determined by the intercept method [7,30]. Then, the relationship between the average grain size and mechanical properties was analysed.

## 3. Results and Discussion

### 3.1. Synthesis of Eucryptite Nanoparticles

Lithium aluminosilicates, such as β-spodumene and eucryptite, have low thermal expansion coefficients, typically 0.2 × 10^−6^ °C^−1^ and −0.9 × 10^−6^ °C^−1^, respectively. However, obtaining them from glass–ceramics is expensive (~1500 °C). The synthesis of eucryptite nanoparticles at lower temperatures (850 °C) can be more economically viable and hence environmentally friendly. One of the synthesis possibilities is the colloidal processing via heterocoagulation. To do this, a study to find out the surface charges of each particle in suspension is necessary in order to optimise the formation of the desired crystalline phase by heterocoagulation.

Figure 3 shows the zeta potential (ζ) of Al_2_O_3_ and SiO_2_ powders as a function of pH. For heterocoagulation to occur, the two types of particles must carry opposite charges (namely ζ ≤ −30 mV for one and ζ ≥ +30 mV for the other) [12]. When the pH is near the point of zero charge (PZC), particles tend to agglomerate and form a rigid structure (gel), resulting in a flocculated and uncontrolled suspension [12].

According to Figure 3, SiO_2_ has an extremely low PZC value below pH = 2. These particles carry negative charges and are stable in the pH range from 5 (ζ < −30 mV) to 8 (ζ < −60 mV). SiO_2_ particles used in this work were stabilised with sodium polyacrylate or ammonia [31]. Because the presence of precursors and stabilising additives in colloidal suspensions affects the surface charge, the zeta potential curve of SiO_2_ in Figure 3 differs from those reported in the literature for wet silica powder or colloidal suspensions stabilised with other polymer additives.

Similar discrepancies were reported by others [32,33]. The PZC of Al_2_O_3_ occurs at pH ≅ 9.5 in Figure 3, which is consistent with the commonly reported values of pH = 8–9 [34]. This high PCZ value means that alumina suspensions are very stable at acidic or neutral pH (e.g., 7.8), as the particles carry significant positive charge and repel each other to overcome the natural London–van der Waals forces [14]. Figure 3 suggests that heterocoagulation between Al_2_O_3_ and SiO_2_ should be carried out at neutral to alkaline pH. Therefore, we synthesised eucryptite particles at pH 6.6 (ζ = +40 mV for SiO_2_ and ζ = –50 mV for Al_2_O_3_).

Acetic acid reacts with Li_2_CO_3_ to form lithium acetate in an aqueous solution (pH = 7). When SiO_2_ and Al_2_O_3_ nanoparticles were added to this solution, OH^−^ ions contributed to increase the pH of the final suspension, facilitating this approach without the need for other additives. Several factors affect the pH, including the hydrolysis equilibrium of Li^+^, hydrolysis equilibrium of acetic acid, hydrolysis equilibrium of functional groups on alumina surface, and adsorption on the silica surface. In combination, these factors maintain a favourable pH for heterocoagulation in the suspension to form the eucryptite phase [13].

Figure 4 shows thermal analysis results of the obtained eucryptite samples. According to the thermogravimetric (TG) curve, the mass loss of around 5% in the range of 30–150 °C was due to the elimination of residual moisture. At 250 °C, there was an exothermic event, possibly related to the combustion of acetic acid [11]. At around 400 °C, an intense exothermic peak was observed together with a marked mass loss of around 30%, and this was attributed to elimination of additives in the colloidal silica suspension. A slight exothermic peak was also observed at 700 °C, which may be related to the precipitation of lithium aluminosilicate crystalline phases [35]. Based on these results, we further carried out heat treatment at 700, 750, 800, 850, and 900 °C to identify the best temperature for forming the eucryptite phase with a lower CTE value.

Figure 5 shows the XRD patterns of eucryptite samples after heating to different temperatures. The crystalline phases were quantified by Rietveld refinement and listed in Table 2. The goodness of fit (GOF) is also listed as an important parameter for XRD refinement. The XRD patterns show characteristic peaks of the β-spodumene and eucryptite crystalline phases, typical members of the lithium aluminosilicate family [36,37]. At 700 °C, 53.4% of Al_2_O_3_ remained unreacted, and all SiO_2_ and Li_2_O were consumed to form lithium metasilicate (Li_2_SiO_3_).

The formation of crystalline eucryptite increased at higher temperatures, reaching a maximum at 850 and 900 °C. Below these temperatures, the main phase formed was β-spodumene. Note that Li_2_SiO_3_ was only observed at 700 °C, since this phase is formed at around 640 °C [4].

In Table 2, eucryptite was quantified using two different ICSD charts that both refer to the hexagonal crystalline system and present a density of 2.35 g/cm³. However, ICSD chart 22014 has the lattice parameters of *a* = 5.2690 Å, *b* = 5.2690 Å, and *c* = 11.1050 Å; whereas ICSD 32595 has *a* = 10.5020 Å, *b* = 10.5020 Å, and *c* = 11.1850 Å. This difference is due to small distortions in the crystal lattice structure during heat treatment.

According to calculation results using the Scherrer equation, the sample treated at 850 °C has an average crystallite size of 84.3 nm. The primary particles consisted of one or several crystallites. This particle size was confirmed by the TEM image in Figure 6.

Figure 6 also shows that the particle size ranged from 24 to 65 nm, and there was agglomeration due to strong inter-particle attraction. Eucryptite particles in this size range have never been obtained before at such a low temperature (850 °C), because both the colloidal synthesis and low heat treatment temperature employed here retarded grain growth. Arcaro et al. [11] obtained glass–ceramic powders from an LZS system (Li_2_O-ZrO_2_-SiO_2_) with an average particle diameter of 35 nm by dry milling in an eccentric mill for 4 h, followed by wet milling in a rotary mill for two days. However, that approach required considerable collision energy and much time.

Nanoparticle powders are expected to possess large specific surface areas. However, BET analysis revealed that the prepared eucryptite particles had a specific surface area of only 3.2 m²/g. This low value is attributed to particle agglomeration (see Figure 6) due to van der Waals forces or strong bonds associated with the formation of early sintering necks during processes such as calcination, fusion, and chemical reactions [38]. It is also supported by the high density of this eucryptite powder (2.4 ± 0.1 g/cm^3^), which is not too far from that of the bulk mineral (2.6 g/cm^3^) [39].

A larger amount of eucryptite is formed at higher temperatures, which is expected to reduce the material’s CTE. This is important in the present study, whose aim is generating compressive residual stresses at the alumina/eucryptite interface to increase the fracture toughness [29]. Table 3 lists the CTE values of samples measured at temperatures from 125 to 450 °C.

For comparison, the alumina matrix has a CTE of 7.1 × 10^−6^ C^−1^ in the same temperature range. The sample prepared at 850 °C has a lower CTE value of 0.4 × 10^−6^ °C^−1^ due to the major eucryptite crystalline phase. This low value is consistent with those reported in the literature [8,40,41,42].

Benavente [43] evaluated lithium aluminosilicate (LAS) ceramics sintered at 1200 °C using different methods and reported the CTE values of 0.6 × 10^−6^ °C^−1^ after 2 h of conventional sintering, −1.86 × 10^−6^ °C^−1^ after 5 min of microwave sintering, and 1.57 × 10^−6^ °C^−1^ after 2 min of spark plasma sintering. Based on the XRD and CTE results, we selected the heat treatment temperature of 850 °C for subsequent experiments.

### 3.2. Properties of Alumina/Eucryptite Composites

There are numerous processing methods available for forming alumina and its composites. Uniaxial pressing and slip casting of suspensions are two of the most commonly used. Slip casting of suspensions has the advantage of making it possible to obtain specimens with a higher green density than pressed specimens. However, a study of the suspension’s rheological behaviour is required in order to obtain a stable suspension with the highest possible solids content. This study is complex and, for this reason, uniaxial pressing of powders may be preferred.

Because eucryptite and alumina have different CTE values, adding nanostructured eucryptite can generate residual compressive stresses at the alumina interface and thereby increase the material’s fracture toughness. Hence, alumina/eucryptite composites were produced via colloidal processing.

First, we evaluated the rheological behaviour of submicron-sized alumina (AL) (Figure 7a) and a mixture of AL and eucryptite nanoparticles (Figure 7b). 

The two suspensions behaved similarly. Ceramic suspensions without sonication (0 min) exhibited thixotropic behaviour with large hysteresis cycles. Such thixotropy is commonly observed in plastic materials and ceramic glazes, usually associated with strong interactions that occur during the orientation of nonequiaxed particles in shear flow [44].

When the suspensions were sonicated, the thixotropy decreased with sonication time and became stabilised at 2 and 3 min. The viscosity curves of AL and 5EU (Figure 8a and Figure 8b, respectively) showed a decrease in viscosity as the shear rate increased. In a concentrated suspension, a relatively low viscosity and acceptable elastic response can only be achieved under optimum particle dispersion. Here, the addition of eucryptite nanoparticles did not significantly influence the viscosity of the alumina suspension. Therefore, a sonication time of 3 min was used before slip casting the specimens.

The thermal behaviour of cast samples during densification by sintering was investigated using the linear shrinkage curve to determine the optimum sintering conditions. The most commonly used method to determine shrinkage is dilatometry. Figure 9a plots the linear expansion as a function of temperature, and Figure 9b plots the first derivative of linear expansion as a function of temperature.

According to Figure 9a, the pure alumina sample (AL) begins to shrink at around 1100 °C, while the 5EU sample begins to do so at around 1025 °C. Hence, adding a phase with a lower melting temperature, especially a lithium-based one, affects the sintering behaviour of alumina. The maximum rate of shrinkage for AL appears to be at approximately 1500 °C. Nevertheless, we chose a higher sintering temperature of 1600 °C in subsequent experiments, because the alumina supplier recommended sintering at 1600 °C for 4 h. In the case of 5EU, the maximum rate of shrinkage is at 1400 °C in Figure 9b and decreases considerably at 1600 °C.

Figure 10 plots the linear expansion of 5EU during heating at a rate of 10 °C/min and subsequent holding at 1400 °C. This sample starts to shrink at approximately 1050 °C. The linear shrinkage is 7.4% after reaching the holding temperature and finally approaches 10.3% at ~60 min holding time, practically at the end of sintering. Based on our dilatometric analysis, we decided to sinter the 5EU composite at the two conditions of 1400 °C/1 h and 1500 °C/35 min, in order to maximise sample densification.

Table 4 lists the true, apparent, and relative densities of cast samples sintered under different conditions. When an additive with a lower melting point is added to the alumina matrix, the formation of a liquid phase during heating is expected to reduce the sintering temperature [4].

However, AL.16 (AL sintered at 1600 °C for 2 h) was denser than the two 5EU samples, possibly due to poorer particle packing of the latter during slip casting, which could reduce the density of the green body. The 5EU sample sintered at 1500 °C/35 min was denser than that at 1400 °C/60 min despite a shorter sintering time. The possible reasons will be discussed below in the context of microstructural analysis. Although the densification decreased in the presence of eucryptite, these samples also required lower sintering temperatures and shorter sintering times than those of pure alumina. Furthermore, compressive stresses originating in the alumina matrix are expected to compensate for the porosity, leading to a higher fracture toughness of the composite.

Next, SEM was used to compare the microstructural features (average size and shape of grains, defects, and porosity) of pure alumina (AL.16) and alumina/eucryptite composites (5EU.14 and 5EU.15). Figure 11 shows the micrographs of samples that had been polished and heat-treated to reveal the grains.

The micrographs of AL.16 (Figure 11a,b) show a well-densified structure with a low apparent porosity (99.0% densification according to Table 4), elongated and coarse alumina grains, and a broad range of grain sizes (1.1–6.5 µm, average value: 2.3 ± 1.2 µm). Arrows in the micrographs indicate the defects (larger pores) produced during sample preparation. In contrast, the sample containing 5 vol.% eucryptite and sintered at 1400 °C for 1 h (5EU.14) consisted of smaller particles (average size 0.9 ± 0.4 µm, Figure 11c,d).

This may be due to the lower sintering temperature (by 200 °C) and shorter holding time (60 min) compared with AL.16. At 5000x magnification, defects ranging in size from 1.5 to 3 µm were observed, together with relatively fewer irregular alumina grains. Alumina grains in the 5EU.15 sample had an average size of 1.2 ± 0.6 µm and irregular shapes (Figure 11e,f), being larger than those in 5EU.14 but smaller than those in AL.16. Therefore, despite a shorter holding time, a higher sintering temperature led to larger alumina grains.

Table 5 compares the measured properties of alumina-based materials, namely the average porosity, average grain size, average Griffith defect size, modulus of elasticity, biaxial flexural strength, hardness, and fracture toughness with those ones found in the literature. The composite identified as A_F_7 incorporated 7 vol.% of a glass–ceramic from the LZSA system in an alumina matrix (grain size: 0.50 µm) and was sintered at 1600 °C for 240 min. The other composite designated as “AL 0.5 7%” contained 7 vol.% of a glass–ceramic from the MgO-SiO_2_-Al_2_O_3_ system in an alumina matrix (grain size: 0.50 µm) and was sintered at 1600 °C for 30 min. Compared to these two composites, our composites containing 5 vol.% eucryptite showed superior elastic modulus, biaxial flexural strength, hardness, and fracture toughness, despite employing sintering temperatures that are 100–200 °C lower. Pure alumina sintered at 1600 °C for 2 h (AL.16) exhibited the best mechanical properties compared to the composites in Table 5. A higher porosity is correlated with lower hardness and biaxial flexural strength.

In Table 5, samples AL.16, 5EU.14, and 5EU.15 show higher hardness values (16.2 ± 0.4, 13.3 ± 0.7, and 12.9 ± 0.4 GPa, respectively) than that of AL 0.5 7% (10.8 ± 0.5 GPa). The 5EU.15 and 5EU.14 samples also display significantly higher biaxial flexural strengths (301 ± 19 and 286 ± 41 MPa, respectively) than those of A_F_7 (273 ± 7 MPa) and AL 0.5 7% (202 ± 39 MPa).

The fracture toughness of 5EU.14 and 5EU.15 were 3.4 ± 0.6 and 3.1 ± 0.5 MPa·m^0.5^, respectively, which were lower than that of A_F_7 (4.27 ± 0.2 MPa·m^0.5^). This difference may be attributed to a lower compressive stress at the interfaces between the secondary phase and alumina matrix in 5EU.14 and 5EU.15. Specifically, A_F_7 featured a much larger average grain size of 15.05 µm, which likely facilitated the distribution of the secondary phase around the grains, generating higher compressive stresses and enhancing the fracture toughness.

The size of natural Griffith defects in alumina ceramics directly influences the material strength. These defects are small cracks or internal defects related to the manufacturing process or the nature of ceramics [45]. While studying the mechanical properties of porcelainized tile compositions using residual stresses, De Noni et al. stated that the size of natural defects directly affects the modulus of elasticity and fracture energy because of the state of microscopic stresses [29]. Residual stresses were formed by the addition of a phase with lower CTE than that of the ceramic matrix.

The Griffith natural defect size was calculated based on the flexural strength and fracture toughness data. Sample AL.16 had the smallest defect size (12.4 µm), while 5EU.15 had a higher flexural strength (301 ± 19 MPa) and smaller defect size (25.6 µm) than sample 5EU.14, which had a natural Griffith defect size of 35.9 µm. However, considering the higher porosity and larger critical Griffith defect in the composite samples 5EU.14 and 5EU.15, it is possible that the reduction in values was not greater owing to the reduction in average grain size observed in relation to AL.16.

The modulus of elasticity of AL.16 was 370 GPa, and the 5EU.14 and 5EU.15 composites had similar values of 350 GPa. Montedo et al. [5] evaluated the effect of adding 5, 10, and 15 vol.% of a glass–ceramic from the LZSA system (Li_2_O-ZrO_2_-SiO_2_-Al_2_O_3_) to three alumina with different particle sizes (0.5, 1.7, and 2.8 µm) followed by conventional pressing and sintering at 1600 °C for 4, 7, and 10 h. In two samples containing 5 vol.% LZSA, the maximum elastic moduli were 290 to 300 GPa and the relative densities were 98.2% and 98.8%, respectively.

Fabris et al. [30] studied the effect of adding a cordierite-based glass–ceramic (MgO-SiO_2_-Al_2_O_3_) to alumina with different particle sizes (0.5 and 1.7 µm), followed by the conventional pressing process. The samples with 7 vol.% of glass–ceramic sintered at 1471 °C/30 min and 1600 °C/30 min both showed more than 99% densification, with the elastic moduli of 350 and 330 GPa and flexural strengths of 205 and 175 GPa, respectively. Suárez et al. [19] studied low-CTE ceramic composites based on SiC-reinforced LAS using a slip casting method. The LAS ceramics were obtained from kaolinite, lithium carbonate, and commercial alumina by grinding in ethanol and heat treatment at 900 °C. Composites containing 75 vol.% LAS and 25 vol.% SiC were sintered at 1430 °C in a non-oxidant atmosphere for 2, 4, and 6 h. The specimens achieved densification of 98.2–99.6%, fracture toughness of 144–162 MPa, hardness of 6.4–6.8 GPa, and modulus of elasticity of 137–150 GPa.

Regardless of the processing method (pressing or slip casting), these previously reported ceramics were inferior in their mechanical properties compared to the composites obtained by suspension processing in this study, even though the latter also showed a higher porosity (possibly due to the smaller average grain size of alumina). In fact, reducing the sintering temperature and holding time leads to a reduction in the alumina grain size [2], which should increase the fracture toughness. The samples sintered at 1400 °C/60 min (5EU.14) showed a fracture toughness of 3.4 ± 0.6 MPa·m^0.5^, which is similar to those of AL.16 and 5EU.15 (3.2 ± 0.3 and 3.1 ± 0.5 MPa·m^0.5^, respectively). Thus, the addition of eucryptite reduces the sintering temperature of alumina and suppresses grain growth [46].

In a previous study, retarded grain growth was observed after adding a glass–ceramic from the LZSA system [7]. In that study, alumina with an initial particle size of 0.5 µm and sintered at 1600 °C for 4 h had a porosity of 13.4%, while adding 5 wt.% of the glass–ceramic reduced the porosity to 1.2%. Micrographs also showed that increasing the number of firing stages was more effective in increasing the grain size than the density. A smaller grain size had a positive effect up to a certain size limit, after which the Hall–Petch mechanism can occur. In this mechanism, the material strength is inversely proportional to the square root of the grain size, that is, a smaller grain size is directly linked to a greater material strength [47]. Thus, we hope that better processing parameters in slip casting suspensions of the studied composite can reduce the internal defects, especially porosity, leading to a higher fracture toughness. In addition, other eucryptite nanoparticle contents should be tested to determine the optimal composition.

## 4. Conclusions

Eucryptite nanoparticles were successfully synthesised through a colloidal process based on heterocoagulation followed by a solid-state reaction for the first time. These nanoparticles were added to an alumina matrix, and ceramic samples were further obtained after conventional slip casting and sintering the composite. The addition of low coefficient of thermal expansion lithium aluminosilicate nanoparticles reduces the sintering temperature of the alumina, suppressing grain growth, and can create residual compressive stresses in the alumina particles. These associated mechanisms can toughen the alumina.

Heat treatment at 850 °C effectively produced nanosized eucryptite particles at 100% yield with a CET value of 0.46 × 10^−6^ °C^−1^, and TEM observations confirmed eucryptite grains in the size range of 16–65 nm. Adding 5 vol.% of the eucryptite nanoparticles to alumina lowered the ceramic sintering temperature by up to 200 °C, achieving 95.3% densification under the sintering conditions of 1400 °C/60 min. This lower sintering temperature resulted in a smaller alumina grain size in all studied samples, leading to a higher fracture toughness compared with alumina composites reinforced with cordierite or spodumene glass–ceramic for the same purpose. In addition, the low CTE value of eucryptite likely generated compressive residual stresses at its interface with alumina, although there is also room for improving this toughening mechanism in terms of porosity and defect formation.

Compared to previous reports on producing ceramic frits at high temperatures (~1500 °C), the approach proposed here is a greener and more economically viable way to produce lithium aluminosilicate nanoparticles. The ceramic composites also exhibited relatively better properties. Nevertheless, the technology is more complex and requires more rigorous process control in nanoparticle production.

## Figures and Tables

**Figure 1 materials-18-00671-f001:**
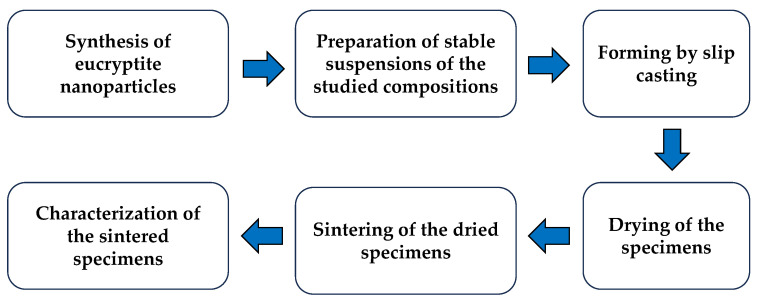
Flowchart of experimental steps.

**Figure 2 materials-18-00671-f002:**
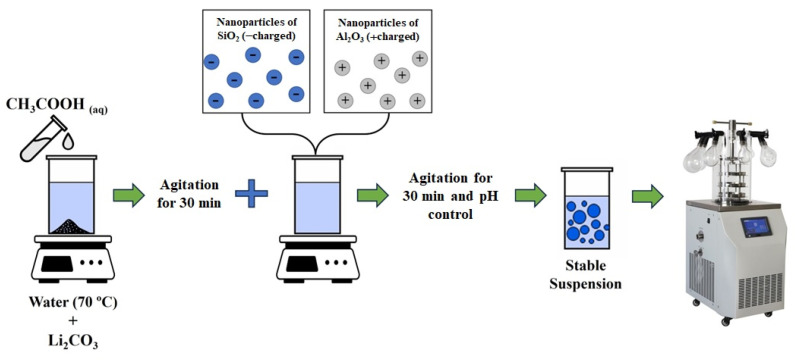
Synthesis of eucryptite nanoparticles by heterocoagulation.

**Figure 3 materials-18-00671-f003:**
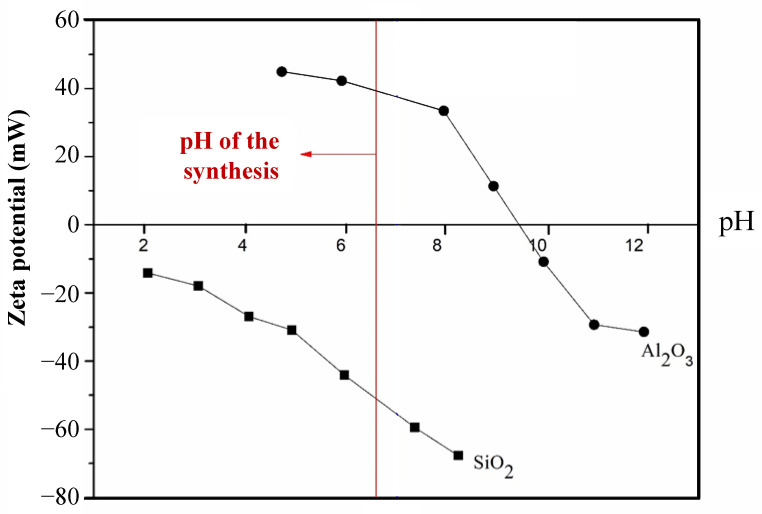
Zeta potentials of silica and alumina as a function of pH. The pH of the eucryptite synthesis is shown in the red line.

**Figure 4 materials-18-00671-f004:**
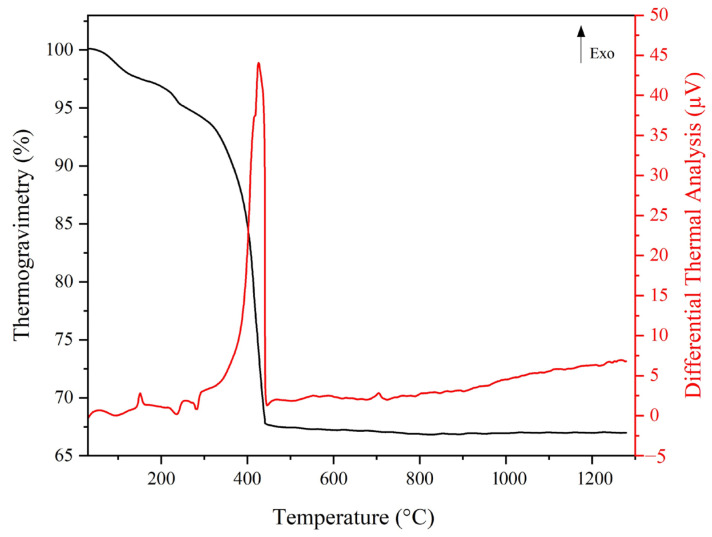
Thermal analysis of as-synthesised eucryptite powder.

**Figure 5 materials-18-00671-f005:**
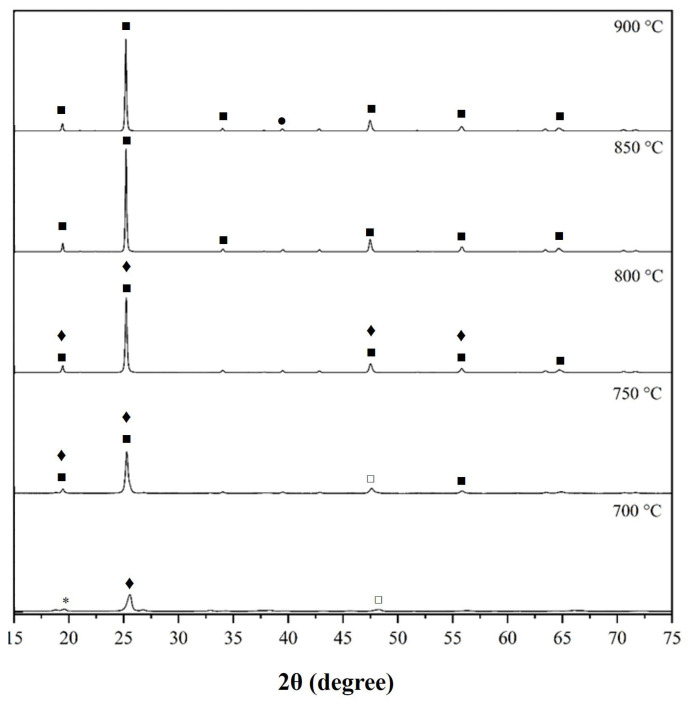
XRD patterns of eucryptite samples prepared at 700–900 °C. ♦: β-Spodumene; ■: Eucryptite; ●: SiO_2_ (low-Quartz); *: Lithium metasilicate; □: Al_2_O_3_.

**Figure 6 materials-18-00671-f006:**
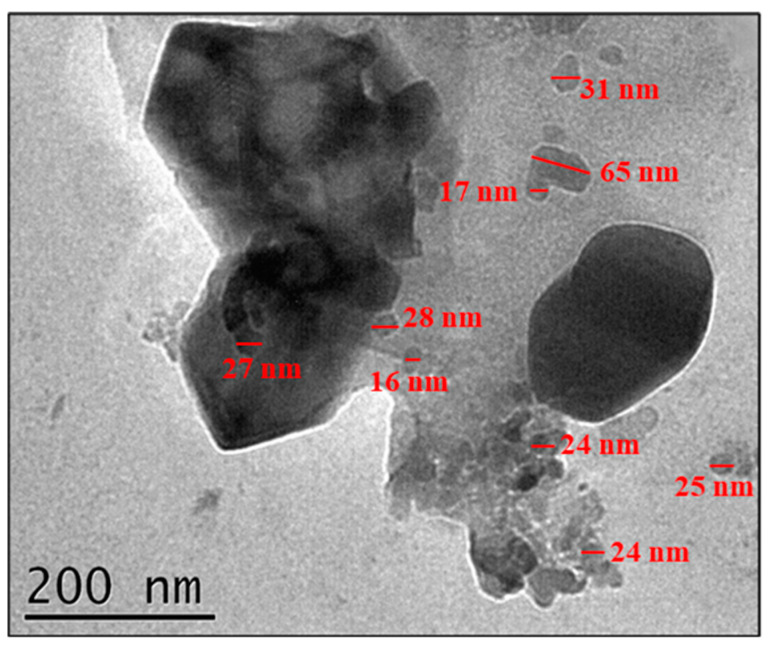
TEM micrograph of eucryptite particles prepared at 850 °C.

**Figure 7 materials-18-00671-f007:**
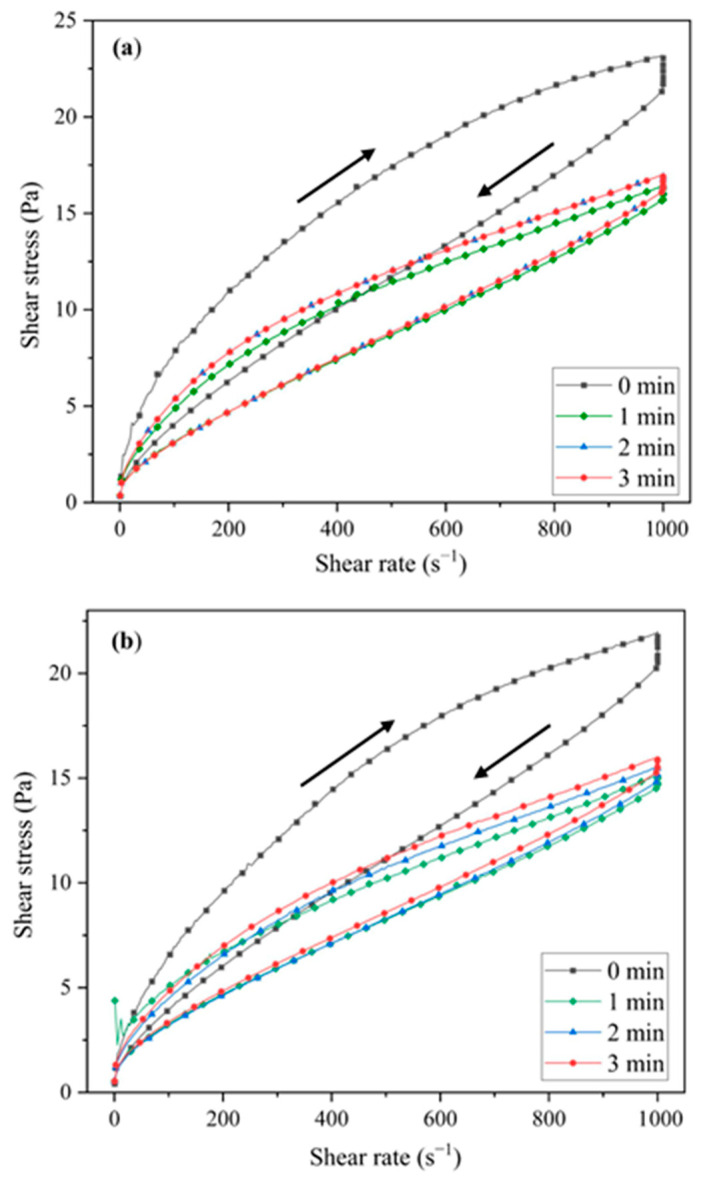
Flow curves of ceramic suspensions (40 vol.% solids) after different sonication times: (**a**) AL and (**b**) 5EU.

**Figure 8 materials-18-00671-f008:**
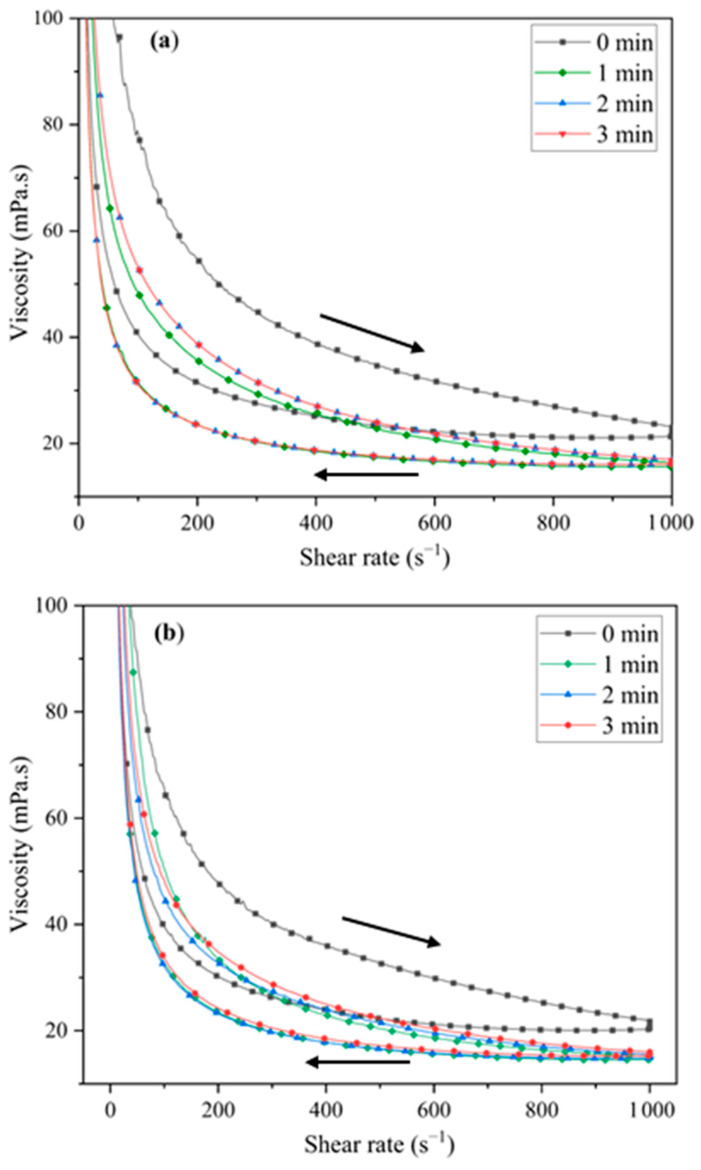
Viscosity curves of (**a**) AL and (**b**) 5EU. Arrows indicate the step of increasing or decreasing shear rate during measurements.

**Figure 9 materials-18-00671-f009:**
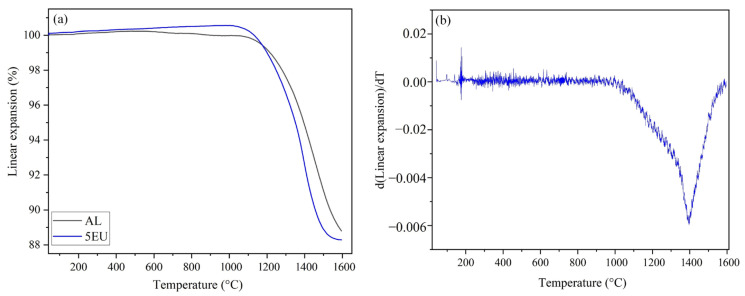
(**a**) Linear expansion curves of AL and 5EU samples during heating at a rate of 10 °C/min up to 1600 °C. (**b**) First derivative of linear expansion curve of composition 5EU.

**Figure 10 materials-18-00671-f010:**
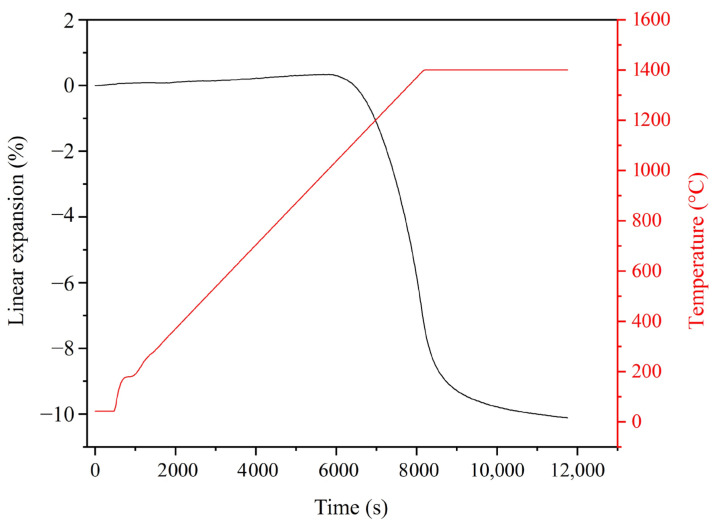
Linear thermal expansion curves of 5EU as a function of time.

**Figure 11 materials-18-00671-f011:**
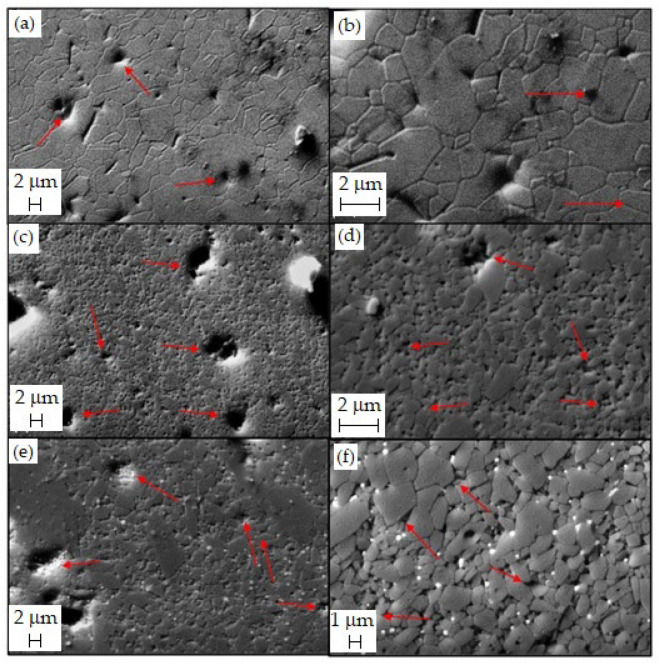
Microstructures of (**a**,**b**) AL.16; (**c**,**d**) 5EU.14; and (**e**,**f**) 5EU.15 samples. Red arrows indicate microstructural defects.

**Table 1 materials-18-00671-t001:** Optimal heat-treatment conditions for each alumina-based composition.

Sample Code	Alumina:Eucriptite Ratio (vol/vol)	T (°C)	Holding Time (min)
AL.16	100:0	1600	120
5EU.14	95:5	1400	60
5EU.15	95:5	1500	35

**Table 2 materials-18-00671-t002:** Quantification of phases by Rietveld method for eucryptite samples prepared at different temperatures.

Temperature (°C)	Content (%)	Crystalline Phase	ICSD Charts	GOF
700	53.4 37.8 8.8	Al_2_O_3_ β-Spodumene Li_2_SiO_3_	9775 24897 853	6.2
750	8.6 42.4 48.9	Al_2_O_3_ Eucryptite β-Spodumene	9775 22014 69221	6.5
800	52.1 47.9	β-Spodumene Eucryptite	69221 22014	5.1
850	65.1 34.9	Eucryptite Eucryptite	22014 32595	4.7
900	99.0	Eucryptite	22011 200723	5.9
	1.0	SiO_2_		

**Table 3 materials-18-00671-t003:** CTE values of different eucryptite samples measured in the range of 125–450 °C.

Preparation Temperature (°C)	CTE (°C^−1^)
700	3.6 × 10^−6^
750	2.3 × 10^−6^
800	1.2 × 10^−6^
850	0.4 × 10^−6^

**Table 4 materials-18-00671-t004:** Apparent density (ρ_ap_), true density (ρ_true_), and relative density (ρ_rel_) of sintered cast samples with and without eucryptite.

Samples	Sintering Temperature (°C)	ρ_true_ (g/cm^3^)	ρ_ap_ (g/cm^3^)	ρ_rel_ (%)
AL	1600	3.94 ± 0.01	3.90	99.0
5EU	1400	3.86 ± 0.00	3.57	92.5
5EU	1500	3.86 ± 0.00	3.60	93.3

**Table 5 materials-18-00671-t005:** Physical and mechanical properties of sintered alumina and various composites.

Property	5EU.14 (1400 °C/ 60 min)	5EU.15 (1500 °C/ 35 min)	A_F_ 7 (1600 °C/ 240 min) [5]	AL 0.5 7% (1600 °C/ 30 min) [30]	AL.16 (1600 °C/120 min)
Porosity (%)	5.6	4.7	5.1	0.74	1.0
Grain size (µm)	0.9 ± 0.4	1.2 ± 0.6	15.05	1.0 ± 0.1	2.3 ± 1.2
Griffith critical flaw size (µm)	35.9	25.6	75	-	12.4
Elastic modulus (GPa)	350	350	290 ± 10	343 ± 4	370
Biaxial bending strength (MPa)	286 ± 41	301 ± 19	273 ± 7	202 ± 39	461 ± 53
Hardness (GPa)	13.3 ± 0.7	12.9 ± 0.4	-	10.8 ± 0.5	16.2 ± 0.4
Fracture toughness (MPa·m^0.5^)	3.4 ± 0.6	3.1 ± 0.5	4.3 ±0.2	2.5 ± 0.2	3.2 ± 0.3

## Data Availability

All the data are available within the manuscript.

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
