# Peer review of "Processing and Characterisation of Alumina/Eucryptite Nanostructured Composites"

_materials, 2025, doi:10.3390/ma18030671_

Round 1
Reviewer 1 Report
Comments and Suggestions for Authors
In this paper, alumina/eucryptite nanostructured composites were prepared and characterized. Some comments need to be addressed before the manuscript is considered for publication.
1) Line 16 “enhance alumina toughness” can be revised as “enhance the toughness of alumina”.
2) Line 22 Please provide mean particle size measured by TEM.
3) Line 24 Does “95.3% densification” mean “relative density of 95.3%”?
4) Line 61 10-6?
5) In the section of “Introduction”, the novelty and motivation of the work needs to be described further.
6) Line 71 an average particle size of 22–24 nm?
7) The scale on the vertical axis in Fig. 4 can be deleted. What is the kind of SiO2?
8) The pixel of some figures is not high. Please provide the original images or photos.
9) Check equation (2).
10) Subscript in symbols such as Ac and Kic.
11) How to evaluate the size of the natural Griffith defect?
12) Line 211 Does “real densities” refer to “true density”? “actual density” in line 360?
13) As mentioned in the manuscript, the formation of pores and defects affected residual pressure. Please explain the influencing mechanism.
14) Please add deviation value to all numerical value in tables if possible.
15) Do not include the magnification of images in the caption of Fig. 9.
16) Due to the different temperature and soaking time of sintering, it’s not easy to compare the data of 5EU and AL16 with each other.
17) In conclusion, the relative density of 95.3% for sample with eucryptite is not high. The basis for the conclusion is not sufficient.
Comments on the Quality of English LanguageThe English could be improved to more clearly express the research.
Author Response
Dear Reviewer,
Enclosed you can find our revised manuscript for your consideration and possible publication in Materials. Please note that we have addressed all the suggestions. We strongly appreciate these suggestions and comments, which we find to be very constructive and helpful and have contributed to strengthen our manuscript. Based on these suggestions and comments, we have made the corresponding modifications in the manuscript, which were highlighted in yellow to facilitate the review process. The detailed response is given in the attached document.

Reviewer 2 Report
Comments and Suggestions for Authors
Dear Authors, the submission Alumina/Eucryptite Nanostructured Composites: Processing and Characterization, Manuscript ID: materials-3408418, has some weaknesses that must be revised appropriately.
Please find below some, of the most significant comments:
1. In the Abstract section, an introduction to the area of study where the presented results can be valuable must be included.
2. Still, referencing the Abstract section, the main advantage of the study presented must be emphasized that it is not obvious what the Authors received.
3. Some general words on the final main advantages should be also provided when reading the Abstract section.
4. From the Introduction section, each of the cited items must be introduced separately, not [2, 4] or [5, 7]. Each of the cited references must be presented with the advantages and disadvantages of the study.
5. There is no critical review in the Introduction section or it is negligible. Only a few sentences addressed some unresolved studies in the previous literature review. Usually, the Authors address some critical aspects to more clearly indicate the motivation of the study.
6. Therefore, in its current form, the motivation included in lines 60-66 does not derive from the lack of the current state of knowledge, which is usually obtained by a critical review of the literature. In the reviewed case, the motivation is not supported by a literature weakness.
7. Considering the 2. Materials and Methods section, the raw materials, except the percentage of their purity, should be presented with their percentage chemical compositions. It must be provided for all of the studied materials.
8. The quality of Figure 1 is poor and does not provide suitable information on the experiment proposed.
9. In section 2 or subsection 2.1, the flow chart of the whole experiment is required. It is difficult to follow what is the main line of the study.
10. It is not clear how were the values from the Table 1 selected. From the current form, it looks like selected arbitrarily.
11. The formula for calculating the Porosity (P) in equation 2 is not newly proposed by the Authors so it should be referenced to the primary source.
12. The advantages of materials containing eucryptite nanoparticles must be highlighted in section 3.2. There is no critical discussion and the limitations are also unknown.
13. Generally, for each of the subsections (3.1 and 3.2) of section 3, the Authors must put some limitations to the study. Usually, it is recommended to address some advantages and disadvantages (limitations) of the analysis or an additional section (The Outlook) is required. Authors can decide if to use more critical discussion or add the mentioned section.
14. The Conclusions must be reworked strongly. Firstly, should be divided into separate and numbered gaps. Secondly, detailed information must be varied from those general. Finally, the main and general conclusion must be included and emphasised at the end of this section.
15. There are many variables and shortcuts that the Nomenclature section is required.
16. Additionally, as a minor comment, the Conclusions section should be section number 4, not 5, that in the current order of the draft, there is no 4th section.
From the above, the reviewed manuscript must be improved significantly before any further processing by the Materials journal, if allowed by the handling Editor.
Author Response

(The authors gave the same response as above.)

Reviewer 3 Report
Comments and Suggestions for Authors
The addition of a second phase with a low coefficient of thermal expansion (CTE) to the alumina matrix can enhance its mechanical properties, reduce the sintering temperature, and increase the toughness by generating compressive stresses on the surface of the alumina particles. In this study, a nanostructured alumina/eucryptite composite was developed to enhance alumina toughness.
- The authors should explain how and why they decided to use Eucryptite nanoparticles in order to improve properties of the alumina.
- 5 vol.% content of eucryptite nanoparticles was defined in this work. The authors should use also other contents of the eucryptite nanoparticles, in order to find optimal amount for the best properties.
- this study used 0.5% of polymeric dispersant (DURAMAX™ D-3005) to stabilize the alumina suspension. What is chemical structure of the polymer. The authors should use also other contents of the dispersant, in order to find optimal amount for the best properties of the product.
- Properties of the new alumina composite reinforced with eucryptite nanoparticles should be compared with those of other alumina composites, which are described in literature. Advantages and disadvantages of the new composite should be described in conclusions.
Author Response

(The authors gave the same response as above.)

Round 2
Reviewer 1 Report
Comments and Suggestions for Authors
This paper has been revised according to comments.
Reviewer 2 Report
Comments and Suggestions for Authors
All of the raised issues were resolved and the submission improved appropiately so the manuscript can be considered for publication in its current form.
Reviewer 3 Report
Comments and Suggestions for Authors
Accept in present form